# The Effect of Vitamin D_3_ Supplementation on Hepcidin, Iron, and IL-6 Responses after a 100 km Ultra-Marathon

**DOI:** 10.3390/ijerph17082962

**Published:** 2020-04-24

**Authors:** Katarzyna Kasprowicz, Wojciech Ratkowski, Wojciech Wołyniec, Mariusz Kaczmarczyk, Konrad Witek, Piotr Żmijewski, Marcin Renke, Zbigniew Jastrzębski, Thomas Rosemann, Pantelis T. Nikolaidis, Beat Knechtle

**Affiliations:** 1Department of Molecular Biology, Gdansk University of Physical Education and Sport, 80-336 Gdansk, Poland; 2Department of Athletics, Gdansk University of Physical Education and Sport, 80-336 Gdansk, Poland; wojciech.ratkowski@awf.gda.pl; 3Department of Occupational, Metabolic and Internal Diseases, Medical University of Gdansk, 81-519 Gdynia, Poland; wolyniecwojtek@gmail.com (W.W.); marcin.renke@gumed.edu.pl (M.R.); 4Department of Clinical and Molecular Biochemistry, Pomeranian Medical University, 70-11 Szczecin, Poland; mkariush@gmail.com; 5Department of Biochemistry, Institute of Sport, National Research Institute, 01-982 Warsaw, Poland; konrad.witek@insp.waw.pl; 6Faculty of Physical Education, Jozef Piłsudski University of Physical Education in Warsaw, 01-813 Warsaw, Poland; piotr.zmijewski@insp.waw.pl; 7Department of Physiology, Gdansk University of Physical Education and Sport, 80-336 Gdansk, Poland; zb.jastrzebski@op.pl; 8Institute of Primary Care, University of Zurich, 8091 Zurich, Switzerland; thomas.rosemann@usz.ch; 9Exercise Physiology Laboratory, 18450 Nikaia, Greece; pademil@hotmail.com; 10Medbase St. Gallen Am Vadianplatz, 9001 St. Gallen, Switzerland

**Keywords:** ultra-endurance, metabolism, nutrition, iron deficiency

## Abstract

Deficiencies in iron and vitamin D are frequently observed in athletes. Therefore, we examined whether different baseline vitamin D_3_ levels have any impact on post-exercise serum hepcidin, IL-6 and iron responses in ultra-marathon runners. In this randomized control trial, the subjects (20 male, amateur runners, mean age 40.75 ± 7.15 years) were divided into two groups: experimental (VD) and control (CON). The VD group received vitamin D_3_ (10,000 UI/day) and the CON group received a placebo for two weeks before the run. Venous blood samples were collected on three occasions—before the run, after the 100 km ultra-marathon and 12 h after the run—to measure iron metabolism indicators, hepcidin, and IL-6 concentration. After two weeks of supplementation, the intervention group demonstrated a higher level of serum 25(OH)D than the CON group (27.82 ± 5.8 ng/mL vs. 20.41 ± 4.67 ng/mL; *p* < 0.05). There were no differences between the groups before and after the run in the circulating hepcidin and IL-6 levels. The decrease in iron concentration immediately after the 100-km ultra-marathon was smaller in the VD group than CON (*p* < 0.05). These data show that various vitamin D_3_ status can affect the post-exercise metabolism of serum iron.

## 1. Introduction

Iron-deficiency anemia in athletes may lead to a decrease in VO_2_ max (the maximum rate of oxygen consumption measured during incremental exercise), an attention deficit and constant fatigue, which, in turn, may affect their athletic performance [1]. Iron metabolism is regulated at both the cellular and the systemic level. At the systemic level, this nutrient is required as a component of haem, including hemoglobin and myoglobin, which are essential for the delivery and storage of oxygen. Iron is also required for other physiological processes that are basic for athletic performance, such as energy production, DNA synthesis and repair, as well as cell division [2]. Conversely, an excess of free iron may be toxic, because the highly reactive atoms, unrelated to protein, react with the reactive oxygen species or lipid peroxides. This property leads to potential toxicity if the concentration of free iron is not properly managed by cells [3].

The main regulator of systemic iron homeostasis is hepcidin, a 25-amino-acid peptide hormone that is produced by hepatocytes [4]. Hepcidin controls serum iron levels by binding to ferroportin (Fpn) leading to its endocytosis and degradation in lysosomes. Through this mechanism, elevated hepcidin prevents iron absorption and recycling, reducing plasma iron and finally body iron stores, whereas decreased hepcidin conversely enhances absorption and recycling, increasing iron in the plasma and raising iron stores [5]. The expression of hepcidin is transcriptionally regulated by iron, erythropoiesis, hypoxia, and inflammation. Extracellular and intracellular iron concentrations increase hepcidin transcription through a mechanism dependent on the bone morphogenetic protein/small mothers against decapentaplegic (BMP)/(SMAD) signaling pathway [6]. Hepcidin production is suppressed during the period of intense iron utilization and reparative or ineffective erythropoiesis, resulting in increased iron delivery to the bone marrow [7]. The pathways involved in hepcidin regulation by erythropoiesis have not been explained yet. Overall, several main candidates have been proposed as erythroid regulators of hepcidin expression: erythropoietin and erythroferrone, growth differentiation factor 15 (GDF15), and twisted gastrulation protein homolog 1 (TWSG1) [8]. The other variable which can influence post-exercise hepcidin levels is hypoxia; nevertheless, the findings that exist are inconsistent, with some reporting an attenuated post-exercise response of serum hepcidin compared to normoxic conditions and others reporting no effects [9,10,11].

Another important regulator of hepcidin is inflammation. During inflammatory states, hepcidin expression is induced by the cytokine IL-6 via the Janus kinase (JAK) signal transducer and activator of transcription (STAT) 3 pathway with synergistic effects from the BMP pathway [12,13,14]. In recent years, many studies have demonstrated that both single bouts and regular exercise cause the rise of blood IL-6 and hepcidin. During and after a 100 km run the regulation of hepcidin and iron metabolism can be controlled with two opposite metabolic pathways: first, the occurrence of inflammation and an increased IL-6 serum concentration which can induce hepcidin synthesis; and second, elevated erythropoiesis activity, which can suppress the elevation of this hormone in serum [15,16]. Thus, although IL-6 is an undoubted co-operator to the post-exercise hepcidin response, it seems that there are additional factors that cooperate during prolonged exercise to achieve the adequate level of this hormone [17].

One of these factors appears to be vitamin D. It is suggested that this vitamin may affect erythropoiesis by a few different pathways, e.g., by its influence on hepcidin synthesis (dependent or independent on cytokines) and by a direct impact on erythroid progenitor cells [18,19].

Several studies found that vitamin D deficiency tends to co-occur with anemia [20,21]. In a Polish population of female athletes, the percentage of iron deficiency subjects was higher (32%) in the vitamin D insufficient group than in the vitamin D sufficient group [22]. In support of the potential function of vitamin D, researchers proposed vitamin D and the hepcidin-ferroportin iron regulatory axis [23]. In their pilot study with healthy volunteers, they observed that elevated serum concentrations of 25(OH)D (but not 1,25 (OH)D) after a single oral dose of vitamin D_2_ (100,000 international units - IU) led to a 34% decrease in serum hepcidin concentration. They suggested that in the case of vitamin D deficiency, elevated concentrations of hepcidin may decrease systemic iron resources. In turn, in the case of vitamin D sufficiency, a decreased concentration of hepcidin may increase systemic levels of iron [23]. A similar impact on serum hepcidin concentrations after vitamin D_3_ treatment (250,000 IU) was also observed by other authors [18]. This effect of vitamin D on the hepcidin-ferroportin axis implies that various vitamin D status may be a contributing factor to the modeling of pre- and post-exercise hepcidin metabolism. Despite the strong evidence for the association between high vitamin D status and the decrease of serum hepcidin concentration, the connection between various vitamin D status and hepcidin, iron and IL-6 responses after an ultra-marathon run has not been investigated. Therefore, this study aimed to examine whether two-week high-dose supplementation (10,000 IU/day) of vitamin D_3_ can have an influence on 25 (OH)D serum concentration, and secondly, whether it can affect hepcidin, iron, and IL-6 responses to a 100-km ultra-marathon. Based on the evidence that both vitamin D_2_ and vitamin D_3_ can lead to decreased serum hepcidin levels, we hypothesized that two-week high-dose vitamin D_3_ supplementation would be effective in increasing serum 25(OH)D levels and reducing post-exercise hepcidin responses, which in turn would lead to an increase in the availability of iron.

## 2. Materials and Methods

This randomized, double-blind, placebo-controlled study was accepted by the Local Ethical Board Committee (approval no. NKBBN/448/2016). One of the most important predictor variables for a successful ultra-marathon performance seems to be previous experience [24]; therefore, athletes with at least three years of ultra-marathon training experience (6.31 ± 7.57 years) were invited to participate in this study. Exclusions criteria included professional training and competing at national or international level (e.g., personal best marathon time <2 h 50 min). The sports experience of study participants has been presented in detail previously [25]. All potential candidates were invited to attend a screening visit where eligibility for inclusion was assessed and informed consent was obtained. Additional exclusion criteria included factors such as 25(OH)D < 20 ng/mL and iron-deficiency anaemia.

Subjects were randomized to two groups—VD (the experimental one, n = 10) and CON (the placebo one, n = 10) with the same age, experience in running and 25(OH)D level: 27.26 ± 7.08 ng/mL and 27.13 ± 3.67 ng/mL, respectively. The VD group received vitamin D_3_ supplementation (around 10,000 UI/day, i.e., 20 droplets—Vigantol medicament, Merck) and the CON group received a placebo (20 droplets of sunflower oil, placed in an identical bottle as vitamin D_3_) for two weeks before a run. The last vitamin D dose was taken 24 h before the 100 km ultra-marathon. Participants were asked not to change their diet or lifestyle during the experiment as well as to refrain from introducing any supplementation treatments. Sunlight exposure was minimal during the study due to the latitude where the athletes were based (54° N).

Twenty male athletes took part in the 100 km ultra-marathon. Seventeen runners completed the study protocol and their results were taken for further statistical analysis. The flow of the inclusion and exclusion of athletes through the study is presented in Figure 1.

The baseline characteristics of both groups, VD and CON, are presented in Table 1.

VO_2_ max was measured with the use of expiratory gas analyser Oxycon Pro (Erich JAEGER GmbH, Hoechberg, Germany, 2012). Experimental runs were performed in physical effort laboratory in standard conditions (temperature 21 °C; atmospheric pressure 1010 hPa; air humidity 55%). The physical tolerance test was preceded by an ergometric work (treadmill Cosmos Saturn, Italy). Bruce’s protocol was used [26]. The highest relative oxygen consumption, maintained for 15 s, at the end of the exercise was considered as VO_2_ max.

### 2.1. Experimental Run

The experiment was conducted at the university stadium (400 m track circumference) in November 2016. Participants were instructed to run a 100 km distance as fast as they could. The trial and weather conditions have been presented in detail previously [25]. Venous blood samples were collected four times: before supplementation, before the run (pre), after 100 km (post) and 12 h after the run (post12). Each venous blood sample was collected in a 3 × 4 mL SST gel separator tube and left 30 min at room temperature for a clot to form (BD Vacutainer Blood Collection system, Becton Dickinson, USA). Serum was separated by centrifuging samples at 1000 G for 10 min. Samples were stored frozen at −80 °C.

### 2.2. Biochemical Analyses

The first part of the study was carried out in the laboratory of the Department of Biochemistry at the Institute of Sport, accredited by the Polish Centre for Accreditation (no. AB 946). The measurements of iron and unsaturated iron-binding capacity (UIBC) were performed using spectrophotometric methods, and measurements of ferritin and soluble transferrin receptor were performed using turbidimetry methods with a Cobas Integra 400 Roche biochemical analyser (Roche Diagnostics, Basel, Switzerland), using original manufacturer reagent kits. Serum levels of 25 (OH)D were measured using a solid-phase ELISA test (DIAsource ImmunoAssays SA, Ottignies-Louvain-la-Neuve, Belgium) and parathyroid hormone (PTH) was measured using electrochemiluminescence (Roche Diagnostics, Basel, Switzerland). Serum calcium was determined using colorimetry (bioMerieux, Craponne, France). Serum phosphate was analyzed using a colorimetric method (Roche Diagnostics, Basel, Switzerland).

The second part of this study was conducted in the Biochemistry Laboratory at the Gdansk University of Physical Education and Sport. Serum hepcidin concentration was measured using a commercially available enzyme-linked immunosorbent assay (ELISA) kit (Wuhan EIAab Science Co., Wuhan, China), according to the manufacturer’s instructions. The intra- and inter-assay coefficients of variation were between 6.9 and 8.9%. The IL-6 concentration was determined using enzyme-linked immunosorbent assay following the manufacturer’s instructions (R&D Systems, Minneapolis, Minnesota, United States). The intra- and inter-assay coefficients of variation were between 4.2% and 6.4%.

### 2.3. Statistical Analyses

Data were analyzed using within-subject modeling (www.sportsci.org). Both pre-supplementing measures and the supplementing responses were compared using a *t*-test, an independent sample *t*-test for unequal variances for pre-supplementing and a dependent sample *t*-test for the responses. In addition, the post- and pre-running difference (a change score) was calculated for each participant and the mean change scores were compared between CON and VD with the *t*-test for unequal variances. To estimate the magnitude of the supplementation effect, the mean change score in non-supplemented individuals was subtracted from the mean change score in supplemented individuals. The difference in mean change score was then standardized with a pre-running standard deviation calculated for all supplemented and non-supplemented individuals according to the method proposed by Hopkins [27]. The differences in mean change and standardized difference were reported with 95% confidence limits. The magnitude of individual responses to training was described by Hopkins [27]. The standard deviation of individual responses (SDIR) was calculated as the square root of the difference between squares of the standard deviations of the change in scores in the supplemented and control groups. The remaining analyses were performed using STATISTICA (StatSoft, Inc., Tulsa, Oklahoma, United States), version 12. A *p*-value < 0.05 was considered significant.

## 3. Results

There were no differences between VD and CON groups in the baseline level of 25 (OH)D. In the vitamin D_3_ supplemented group, the 25(OH)D level remained stable from 27.26 ± 7.09 ng/mL before to 27.82 ± 5.88 ng/mL after the supplementation, whereas in the CON group, the level decreased significantly from 27.13 ± 3.67 ng/mL to 20.41 ± 4.67 ng/mL (*p* < 0.001).

In parallel with the analysis of 25(OH)D, the assays were also carried out to estimate circulating concentrations of indicators classically associated with its function. No significant differences were found between the two groups in serum PTH concentrations (VD 37.52 ± 15.08 pg/mL vs. CON 42.2 ± 7.53 pg/mL). Likewise, there were no significant changes in serum calcium concentration (VD 9.98 ± 0.4 mg/dL vs. CON 9.72 ± 0.46 mg/dL) or in serum concentration of phosphate (VD 3.56 ± 0.37 mg/dL vs. CON 3.66 ± 0.61 mg/dL). Furthermore, there were no differences between the groups before the run (pre) in the circulating hepcidin, serum iron concentration, iron metabolism associated parameters and IL-6 levels (Table 2).

Analysis of circulating hepcidin levels showed a significant increase after 100 km (post) compared to the baseline (pre) in both groups combined (pre 0.58 ± 0.25 vs. after 1.04 ± 0.79 *p* = 0.049, effect size = 0.61). However, this increase was not observed for all runners (in two subjects in the VD group, in three in the CON group). Furthermore, we found statistically significant differences between the groups in the post-exercise iron responses (post) (Table 2). Further analyses showed no statistically significant changes between VD and CON groups (Table 3).

## 4. Discussion

This study investigated, initially, whether the two-week high-dose supplementation (10,000 IU/day) of vitamin D_3_ can have an influence on 25 (OH)D serum concentration and, secondly, whether it can affect hepcidin, iron and IL-6 responses to a 100-km ultra-marathon.

As expected, after two weeks of the supplementation, we observed that the intervention group demonstrated a higher level of serum 25(OH)D than the CON group (27.82 ± 5.8 ng/mL vs. 20.41 ± 4.67 ng/mL; *p* < 0.05), Interestingly, contrary to our hypothesis, in the present study, there were no significant differences in exercise-induced responses of hepcidin and IL-6 after the vitamin D_3_ treatment. Nevertheless, we observed statistically significant differences between the groups in post-exercise iron responses (post).

Recent evidence showed that vitamin D can affect physical performance through remodeling and improvement of muscle function, muscle regeneration, maintaining bone health, and a reduction of risk of infections [28,29,30,31]. Some studies have also demonstrated that vitamin D affects athletic performance [32,33,34,35,36]. Therefore, the interest in vitamin D supplementation is increasing in response to these observations and many athletes are now engaged in vitamin D supplementation, using the varied timing and dosage of supplementation [37].

While case reports are limited, vitamin D toxicity after supplementing with exogenous vitamin D has also been noted in amateur athletes. However, these side effects have been observed after using extreme doses (even 35,000,000 IU/100 mL in intramuscular injection) and after a long time [38,39]. Owens et al. reported that dosing strategies for vitamin D in professional sports generally range from 1000 IU per day to blanket supplementation of up to a 100,000 IU dose per week [40]. The data presented in this study suggest that supplementation of vitamin D_3_ 10,000 IU/day for two weeks can be considered to maintain a stable 25(OH)D, although the two-week period seems to be ineffective to achieve a sufficient 25(OH)D level >30 ng/mL. In a randomized controlled trial, our group showed that vitamin D_3_ supplementation with 5000 IU for eight weeks increased the 25(OH)D level to the cut-off of >30 ng/mL [41]. Moreover, a recent study provided a reflection on high-dose vitamin D supplementation. In this randomized clinical trial, the treatment with vitamin D_3_ for 3 years at a dose of 10,000 IU/day and 4000 IU/day resulted in a decrease in radial bone mineral density whereas the treatment with the daily dose of 10,000 IU/day caused a decline in tibia mineral density too [39]. Therefore, the lower than 4000 IU daily vitamin D_3_ dose seems to be more effective and safer to achieve sufficiency in 25(OH)D concentration, in latitudes where there is little sun exposure in both autumn and winter.

As expected, we observed that the level of 25(OH)D decreased in the CON group after the two autumn weeks. This statistically significant drop was fast and large (from 27.13 ± 3.67 ng/mL to 20.41 ± 4.67 ng/mL). In contrast to the current study, Sawyer et al. showed only a moderate decline of vitamin D level after four weeks [42]. On the other hand, Schoenmakers et al. found in their mathematical model that the 25 (OH)D can also decrease fast [43]. Their results are similar to those observed in our study; thus, within a short period, we can observe a fast decrease in 25(OH)D concentration. Although both mathematical models may be helpful to estimate the mean 25 (OH)D level in the general population, it seems that their use among groups of athletes may be problematic.

Previous studies have shown that various vitamin D status may be a contributing factor to the modeling of hepcidin metabolism [18,23]. In our study, two weeks supplementation with 10,000 IU of vitamin D_3_ prevented the reduction of serum 25(OH)D concentrations. Although the differences in 25(OH)D levels between two groups were significant, we did not notice a dissimilarity in hepcidin levels between VD and CON before the run (pre). While few studies have investigated the association of vitamin D with hepcidin, our findings are consistent with the data of a study conducted by Braithwaite et al. [44] where pregnant women received 1000 IU of vitamin D_3_ per day for 14 weeks. Likewise, in older Mexican adults, circulating 25(OH)D concentrations were not associated with hepcidin levels [45]. On the other hand, a double-blind, placebo-controlled pilot trial, where healthy adults received 250,000 IU of vitamin D_3_, showed a 73% decrease in plasma hepcidin concentration [18]. A previous study in healthy volunteers supported the effect of single high doses of vitamin D_2_ on hepcidin synthesis [23]. A major difference between the current study and studies by Bacchetta and Smith et al. is the dose of vitamin D [18,23]. A one-time oral dose of 100,000 IU of vitamin D_2_ and 250,000 IU of vitamin D_3_ can significantly reduce plasma hepcidin concentration (33% reduction vs. 73%) [18,23]. Nevertheless, the Endocrine Society set the limit at 10,000 IU per day [46]. Therefore, the purpose of this study was to examine the effect of the highest recommended dose, 10,000 IU, on plasma 25(OH)D concentration. Consequently, during the two-week period participants received 140,000 IU of vitamin D_3_, although not as a one-time dose. Therefore, it is also possible that the change in hepcidin with the single mega dose of vitamin D was a toxic effect, and that the lack of change with the lower daily doses is a true null effect.

Further, in our study, serum 25(OH)D concentration was 27.82 ± 5.88 ng/mL after supplementation whereas in the trial conducted by Bacchetta et al. the value was 43.5 ± 3 ng/mL [23]. Based on data presented in this study and also in the trial conducted by Braithwaite et al., we suppose that vitamin D_3_ doses from 1000 IU/day to 10,000 IU/day are too small to increase 25(OH)D concentration to a sufficient level which could support inhibition of the synthesis of hepcidin at rest [44].

Generally, physical exercises induce an increase in serum hepcidin concentrations during the early recovery phase [17]. In our study, the post-exercise hepcidin results (post) are consistent with earlier studies reporting significant changes in marathon runners [47]. Roecker et al. concluded that there were two types of hepcidin responses: athletes with peak hepcidin values at least four times higher than pre-race values were designated as responders (n = 8) and the others as non-responders (n = 6) [47]. In contrast to present data in our previous ultra-marathon study, we found that blood hepcidin levels did not significantly change after a 100-km ultra-marathon, although it was a small group of athletes, probably mostly non-responders (n = 6) [15].

Despite the lack of differences in hepcidin levels between VD and CON after the run (post), we observed a treatment-related effect of vitamin D_3_ on changes in serum iron concentrations in response to the 100-km ultra-marathon (post). In the VD group, the drop in serum iron levels was smaller compared to the CON group, which, in turn, may have an effect on the bioavailability of iron during intensified erythropoiesis. These findings suggest that vitamin D may influence post-exercise iron concentrations even without greatly altering hepcidin concentrations. As described earlier, several studies have indicated an association between a deficit of vitamin D and an increased risk of anemia [20,21,22]. However, the mechanism by which vitamin D may affect iron regulation and erythropoiesis remained elusive.

In the present study, the concentration of IL-6 was similar in both groups CON and VD after the two-week supplementation, before the run (pre). This is in contrast to the study of Adegoke et al., where it was shown that daily supplementation of 2000 IU of vitamin D for three months was able to reduce serum concentrations of cytokine IL-6 in children with sickle cell disease [48]. On the other hand, Parsaie et al. observed that supplementation of 50,000 IU of vitamin D/week for eight weeks increased the IL-6 levels in soccer players [49]. These inconsistent findings can be observed due to inflammation caused in different mechanisms, for example, inflammation induced by exercise and inflammatory disease conditions [50].

Despite the significant differences in 25(OH)D between VD and CON groups after two weeks supplementation, before the run, circulating hepcidin and IL-6 concentrations were similar in response to the 100-km ultra-marathon. Likewise, the findings of Dahlquist et al. showed no significant differences in post-exercise IL-6 and hepcidin responses after vitamin K_2_ with vitamin D_3_ supplementation [51]. During physical exercises, the contracting muscles secrete significant amounts of IL-6, which has been proposed to have an anti-inflammatory effect [50]. Our findings do not support the assumption that vitamin D_3_ supplementation could influence the IL-6 serum concentration and its responses after an acute bout of intense exercise.

Finally, the runner’s diet may be discussed as a potential variable, especially an influence of CHO (carbohydrate) ingestion. Although Badenhorst et al. noticed that the timing of CHO ingestion after interval exercise did not affect exercise-induced increases in IL-6 and hepcidin [52]. Furthermore McKay et al. recently showed that despite diet-induced differences in IL-6 response to exercise, post-exercise hepcidin concentrations were similar between diets and trials, indicating CHO availability has minimal impact on post-exercise iron metabolism [53].

To the best of our knowledge, this is the first study aimed at investigating the connection between various vitamin D_3_ status and serum hepcidin, iron and IL-6 responses to a 100-km ultra-marathon. The results of our study support previous observations on the association between iron and vitamin D metabolism. A limitation of the present study was the relatively small group size. However, because of the growing interest in long-distance running among the general population, our study has many strengths, including the examination of extreme effort (~10 h) and amateur athletes.

## 5. Conclusions

Supplementation with the maximal permissible dose of 10,000 IU vitamin D_3_ per day for two weeks had an impact on baseline 25(OH)D concentration but showed no effect on post-exercise serum hepcidin concentration, though it influenced post-exercise serum iron metabolism. For these reasons, vitamin D could be important in the prevention strategies against iron deficiency anemia in endurance athletes, although the use of high doses of vitamin D for this purpose has not been justified. To conclude, both an excess and a deficiency of vitamin D can be detrimental to the selected health indices; therefore, future studies are needed to explore the association between the various doses of vitamin D_3_ supplementation and iron metabolism in athletes, particularly in endurance athletes, who are in a high-risk group for iron deficiency.

## Figures and Tables

**Figure 1 ijerph-17-02962-f001:**
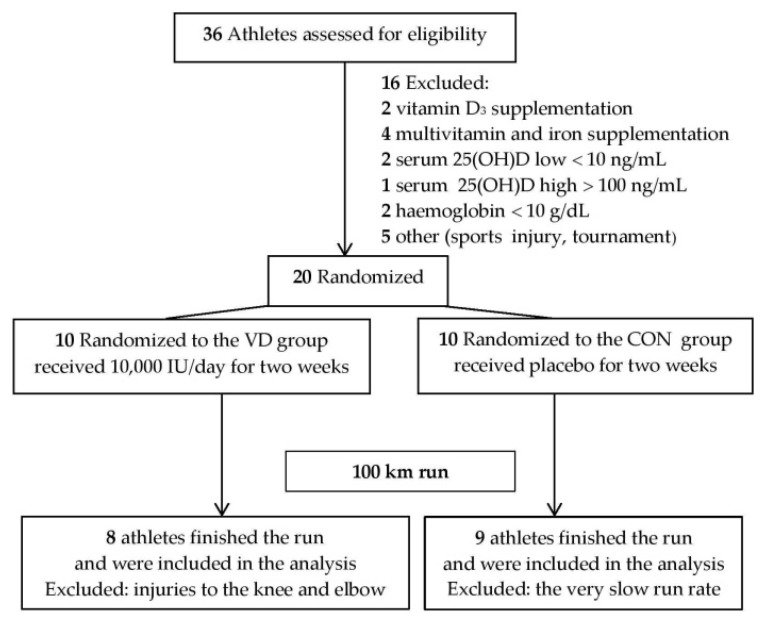
Flow diagram of inclusion and exclusion of athletes through the study. VD—supplemented with vitamin D_3_; CON—placebo group; IU—international units.

**Table 1 ijerph-17-02962-t001:** Comparison of subjects’ baseline characteristics (mean ± standard deviation).

Variable	Vitamin D Group	Control Group	*p*
Number of subjects	10	10	
Age (years)	39.0 ± 5.93	42.5 ± 8.37	0.143
Vitamin D_3_ supplementation	10,000 IU/14 days	0	
Finishers	8	9	
Age (years)	39.56 ± 6.0	40.87 ± 2.36	0.281
25(OH)D—before supplementation (ng/mL)	27.26 ± 7.09	27.13 ± 3.67	0.518
BMI (kg/m^2^)	24.31 ± 2.38	24.21 ± 2.51	0.533
Height (cm)	177.59 ± 6.12	178.59 ± 5.49	0.362
Weight (kg)	76.68 ± 7.93	76.80 ± 8.21	0.488
VO_2_ max (mL × kg^−1^ × min^−1^)	56.65 ± 4.88	53.30 ± 7.34	0.865

VD—supplemented with vitamin D_3_; CON—placebo group; BMI—body mass index; VO_2_ max—the maximum rate of oxygen consumption measured during incremental exercise.

**Table 2 ijerph-17-02962-t002:** The effect of vitamin D_3_ supplementation on exercise-induced changes in parameters of iron metabolism before the run (pre) and after the 100-km ultra-marathon (post).

Variable	Mean ± SD (pre)		*p* ^a^	Mean ± SD (post)		*p* ^a^	Mean Change ± SD		*p* ^a^	Difference in Mean Change (Standardized)	Individual Response as SD
	CON (n = 9)	VD (n = 8)		CON (n = 9)	VD (n = 8)		CON (n = 9)	VD (n = 8)			
Iron (µg/dL)	112.78 ± 50.13	94.14 ± 23.98	0.828	35.78 ± 11.70	66.86 ± 36.24	0.028	−77.00 ± 48.68	−27.29 ± 39.57	0.041	49.7; 2.4; 97.0 (1.26; 0.06; 2.45)	−28.3 (−61.0; 46.0)
Hepcidin (ng/mL)	0.54 ± 0.28	0.63 ± 0.22	0.232	0.72 ± 0.33	1.41 ± 1.02	0.056	0.18 ± 0.47	0.77 ± 0.92	0.196	0.59; −0.39; 1.58 (−0.88; −0.57; −2.34)	0.79 (−0.67; 1.31)
IL6 (pg/mL)	1.04 ± 0.35	0.87 ± 0.31	0.849	23.0 ± 1.40	21.83 ± 2.68	0.851	21.96 ± 1.56	20.96 ± 2.65	0.409	−1.0; −3.6; 1.6 (−3.02; −10.91; 4.87)	2.1 (−2.0; 3.6)
Ferritin (ng/mL)	61.89 ± 34.61	35.43 ± 25.34	0.949	81.11 ± 43.86	47.00 ± 28.68	0.959	19.22 ± 18.48	11.57 ± 9.41	0.303	−7.7; −23.1; 7.8 (−0.24; −0.73; 0.25)	−15.9 (−24.5; 9.8)
UIBC (µg/dL)	199.75 ± 46.56	242.14 ± 60.85	0.069	274.75 ± 36.41	287.14 ± 71.95	0.334	75.00 ± 42.15	45.00 ± 40.27	0.183	−30.0; −76.2; 16.2 (−0.55; −1.40; 0.30)	−12.4 (−52.6; 49.6)
TIBC (µg/dL)	312.13 ± 45.98	336.29 ± 64.97	0.201	309.13 ± 34.14	354.00 ± 58.59	0.051	−3.00 ± 29.73	17.71 ± 36.42	0.256	20.7, −17.3; 58.7 (0.39, −0.32; 1.09)	21.0 (−36.3; 47.0)
sTfR (µg/mL)	3.17 ± 0.75	3.19 ± 0.98	0.482	3.38 ± 1.03	3.07 ± 0.89	0.737	0.21 ± 0.38	−0.11 ± 0.41	0.127	−0.3; −0.8; 0.1 (−0.40; −0.94; 0.13)	0.14 (−0.47; 0.50)

^a^ The *t*-test for unequal variances; CON—placebo group; VD—supplemented with vitamin D_3_; for the difference in mean change and standardized (with pre-running SD) difference in mean, 95% confidence limits are shown. UIBC—unsaturated iron-binding, TIBC—total iron-binding capacity, sTfR—soluble transferrin receptor.

**Table 3 ijerph-17-02962-t003:** The effect of vitamin D_3_ supplementation on exercise-induced changes in parameters of iron metabolism after the 100-km ultra-marathon (post) and 12 h after the run (post12).

Variable	Mean ± SD (post)		*p* ^a^	Mean ± SD (post12)		*p* ^a^	Mean Change ± SD		*p* ^a^	Difference in Mean Change (Standardized)	Individual Response as SD
	CON (n = 9)	VD (n = 8)		CON (n = 9)	VD (n = 8)		CON (n = 9)	VD (n = 8)			
Iron (µg/dL)	37.75 ± 10.79	66.86 ± 36.24	0.033	197.13 ± 69.07	184.29 ± 76.57	0.641	159.38 ± 68.62	117.43 ± 48.12	0.191	−41.9; −107.9; 24.0 (−1.44; −3.70; 0.82)	−48.9 (−89.3; 56.5)
Hepcidin (ng/mL)	0.77 ± 0.33	1.41 ± 1.02	0.068	0.71 ± 0.19	0.78 ± 0.37	0.320	−0.06 ± 0.36	−0.63±0.98	0.229	−0.6; −1.6; 0.5 (−0.72; −2.05; 0.60)	0.91 (−0.61;1.43)
IL6 (pg/mL)	23.00 ± 1.40	21.83 ± 2.68	0.851	11.49 ± 6.83	7.09 ± 4.66	0.931	−11.50 ± 7.82	−14.73±5.37	0.388	−3.2; −11.2; 4.7 (−1.51; −5.22; 2.21)	−5.7 (−10.4; 6.6)
Ferritin (ng/mL)	72.75 ± 38.46	47.00 ± 28.68	0.924	105.63 ± 43.02	71.86 ± 33.38	0.958	32.88 ± 18.97	24.86 ± 9.99	0.320	−8.0; −25.1; 9.1 (−0.20; −0.62; 0.22)	−16.1 (−25.6; 11.6)
UIBC (µg/dL)	286.86 ± 32.43	287.14 ± 71.95	0.496	129.29 ± 43.87	133.86 ± 78.14	0.444	−157.57 ± 45.21	−153.29 ± 62.57	0.886	4.3; −60.6; 69.2 (0.08; −1.15; 1.32)	43.3 (−55.9; 82.9)
TIBC (µg/dL)	325.57 ± 23.85	354 ± 58.59	0.109	309.43 ± 26.01	318.14 ± 52.63	0.339	−16.14 ± 26.81	−35.86 ± 39.84	0.302	−19.7; −60.1; 20.6 (−0.40; −1.22; 0.42)	29.5 (−33.2; 53.3)
sTfR (µg/mL)	3.49 ± 1.04	3.07 ± 0.89	0.803	2.93 ± 0.80	2.74 ± 0.49	0.714	−0.56 ± 0.34	−0.33 ± 0.52	0.334	0.2; −0.3; 0.7 (0.25; −0.29; 0.79)	0.40 (−0.42; 0.70)

^a^ The *t*-test for unequal variances; CON—placebo group; VD—supplemented with vitamin D_3_; for difference in mean change and standardized (with pre-running SD) difference in mean, 95% confidence limits are shown. UIBC—unsaturated iron-binding, TIBC—total iron-binding capacity, sTfR—soluble transferrin receptor.

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
