# Peer review of "The Effect of Vitamin D3 Supplementation on Hepcidin, Iron, and IL-6 Responses after a 100 km Ultra-Marathon"

_ijerph, 2020, doi:10.3390/ijerph17082962_

Round 1
Reviewer 1 Report
This revised manuscript by Kasprowicz et al. focused on the relation among vitamin D, hepcidin, iron, and IL-6. Authors responded and revised the manuscript appropriately according to the reviewer’s comments. It appeared to be better.
Author Response
Response: Thank you very much for your helpful comments.
Reviewer 2 Report
The authors of paper “The Effect of Vitamin D3 Supplementation on Hepcidin, Iron and IL-6 Responses after a 100 km Ultra-marathon”, have studied the effect of supplementation of Vitamin D for two weeks in subjects before the 100 Km ultramarathon. The authors findings that the supplementation did not give influence on IL-6 and hepcidin after run but influence post exercise iron concentrations so one results is that vitamin D may affect iron regulation and erythropoiesis.
one osservation about the table:
TABLE: The Data of table 2 post and data of table 3 post I think could be with the same number…I do not understand the reason that some of these numbers are different
Author Response
Comments and Suggestions for Authors
The authors of paper “The Effect of Vitamin D3 Supplementation on Hepcidin, Iron and IL-6 Responses after a 100 km Ultra-marathon”, have studied the effect of supplementation of Vitamin D for two weeks in subjects before the 100 Km ultramarathon. The authors findings that the supplementation did not give influence on IL-6 and hepcidin after run but influence post exercise iron concentrations so one results is that vitamin D may affect iron regulation and erythropoiesis.
one osservation about the table:
TABLE: The Data of table 2 post and data of table 3 post I think could be with the same number…I do not understand the reason that some of these numbers are different
Response: The data presented in table 2 and 3 are paired (individuals were sampled more than once). The number of pairs pre and post (table 2) were not the same like post and 12 post (table 3). In the table 3 doubtful data was excluded because of haemolyzed blood from one participant 12 hour after the run. Therefore, some of these numbers are slightly different. We want to thank you again for your all helpful comments.
Reviewer 3 Report
- In the Introduction, there should be several sentences regarding hepcidin role per se on the iron metabolism. If so, it would be useful, for the readers that are vitamin D experts but not as such on iron metabolism, to understand better the findings and the imptact of the paper.
- The two groups - vit. D group and the control group - are of very small sample size. Therefore, the use of parametric statistical methods like t-test is a bad choice. Please make calculations using the nonparametric methods, i.e. Mann–Whitney U test, and present the results. Also, not the mean±SD should be reported, but the median and the IQR.
- There is a bad format of text, i.e., the changes of the copyedit were left (Tracking Changes was "on"?). This produces some difficulties in reading and evaluating the text. Please provide a "clear" text, including with the well-formatted reference list.
Author Response
Comments and Suggestions for Authors
- In the Introduction, there should be several sentences regarding hepcidin role per se on the iron metabolism. If so, it would be useful, for the readers that are vitamin D experts but not as such on iron metabolism, to understand better the findings and the imptact of the paper.
Response: As per request the information was added:” Hepcidin controls serum iron levels by binding to ferroportin (Fpn) leading to its endocytosis and degradation in lysosomes. Through this mechanism, elevated hepcidin prevents iron absorption and recycling, reducing plasma iron and finally body iron stores, whereas decreased hepcidin conversely enhances absorption and recycling, increasing iron in the plasma and raising iron stores [5].” Line 57-61
- The two groups - vit. D group and the control group - are of very small sample size. Therefore, the use of parametric statistical methods like t-test is a bad choice. Please make calculations using the nonparametric methods, i.e. Mann–Whitney U test, and present the results. Also, not the mean±SD should be reported, but the median and the IQR.
Response: Basically, Mann-Whiteny U test is not the right choice as the data are paired (individuals were sampled more than once). In that case Wilcoxon signed rank test would be probably more appropriate. However, the choice between parametric and non-parametric test in case of relatively small sample size is not clear-cut. It is almost always arbitrary as there is no sample size above which the parametric tests are allowed to be used in place of non-parametric ones. Certainly, we could compare the changes using Mann-Whitney test, but then we could not analyse the individual responses. The individual responses represent very important aspects of training response/adaptation. The measure of individual responses are based on population/distribution -derived parameters (the same as parametric tests), specifically on standard deviation, thus the whole analysis would be inconsistent. Therefore, we decided to use parametric t-tests for the pre-post differences assuming different variance. Such assumption is reasonable when one anticipate a different treatment response and the presence of individual response. As stated above, we do not insist on doing parametric statistics, as we believe both approaches could be applied here. However, if the reviewer believes that only non-parametric tests are valid in this case, we can carry out non-parametric tests, although some important and valuable results such as individual responses will have to be removed for the sake of consistency.
- There is a bad format of text, i.e., the changes of the copyedit were left (Tracking Changes was "on"?). This produces some difficulties in reading and evaluating the text. Please provide a "clear" text, including with the well-formatted reference list.
Reasponse: To create this manuscript we used the Microsoft World template: https://www.mdpi.com/journal/ijerph/instructions
The tracking changes was on, although we see that some of new changes are accepted, probably by accident by one of the co-authors. We corrected it. The changes of the copyedit are on the right on our manuscript, it may be some technical problem with the World version, although we do not know. Reference list was prepared with a bibliography program, although it contained a few mistakes – thank you for your perceptiveness. The mistakes were corrected.
Round 2
Reviewer 3 Report
Thanks for the corrected manuscript. It is more clear now.
Regarding the statistical methods that were or should have been used, I'd like to point out that for choosing the right statistical test for comparing sub-samples, one of the most important things to be done at first is to check the sub-samples for normality assumption of distribution. It is estimated that for small samples - having less than 25 (some say - less than 20) of cases - such tests for normality like Kolmogorov-Smirnof's test should not be applied at all, event if the distribution of results seems to be "normal" (on the histogram or so). Therefore, the parametric tests are also not to be used for comparisons of such samples at all!
But, from the medical point of view, it is also important, in the process of data interpretation, to clarify whether, simply, there are or there are not differences between certain parameters of the groups, regardless of the statistical significance. If there are no clear differences (in numbers, like the mean or the median), the statistical significance is not so important - since there are no "clinically significant" differences or changes, like there was in the present study, where only some changes were found... So, returning to the manuscript: I could agree that the current presentation of the data (mean ± SD) in the tables be left in the text :)
Author Response
Thanks for the corrected manuscript. It is more clear now.
Regarding the statistical methods that were or should have been used, I'd like to point out that for choosing the right statistical test for comparing sub-samples, one of the most important things to be done at first is to check the sub-samples for normality assumption of distribution. It is estimated that for small samples - having less than 25 (some say - less than 20) of cases - such tests for normality like Kolmogorov-Smirnof's test should not be applied at all, event if the distribution of results seems to be "normal" (on the histogram or so). Therefore, the parametric tests are also not to be used for comparisons of such samples at all!
But, from the medical point of view, it is also important, in the process of data interpretation, to clarify whether, simply, there are or there are not differences between certain parameters of the groups, regardless of the statistical significance. If there are no clear differences (in numbers, like the mean or the median), the statistical significance is not so important - since there are no "clinically significant" differences or changes, like there was in the present study, where only some changes were found... So, returning to the manuscript: I could agree that the current presentation of the data (mean ± SD) in the tables be left in the text :)
Response: We want to express our genuine gratitude for your contributions and support during this manuscript preparation. Your comments have given us a deeper reflection on our study presented in this article especially statistical methods which have been used. As Carl G. Jung said: “Mistakes are, after all, the foundations of truth, and if a man does not know what a thing is, it is at least an increase in knowledge if he knows what it is not.” Thank you again for the inspiring discussion.
This manuscript is a resubmission of an earlier submission. The following is a list of the peer review reports and author responses from that submission.
Round 1
Reviewer 1 Report
The purpose of this study was to examine the effect of two weeks of vitamin D3 supplementation on 25(OH)D serum concentration and its effect on iron, hepcidin and IL-6 responses after a 100 km ultra-marathon. Specific comments and interrogations are listed below.
Abstract
- Line 28, second sentence. What is the purpose of this sentence? Unless there is a rationale, it should be removed.
Introduction
- The introduction is unnecessarily long and places too much emphasis on the regulatory physiology of iron metabolism and hepcidin. Indeed, this is not a mechanistic study and all this information makes us believe that this could be the case. Hence, only basic information necessary for the demonstration of a link between vitamin D, hepcidin, iron metabolism, IL-6 and inflammation should be reported.
- Line 89. What is 25 vitamin D?
- Line 98. Please indicate whether it has been studied for other forms of exercise?
- Line 100. Please explain why measure 25 (OH) D serum concentration?
- Line 104. Vitamin D3 should be replaced by 25 (OH) D.
Methods
- Line 108. Performance was not a primary outcome in this study. Therefore, why were athletes with less experience excluded?
- Line 111 – Why were elite athletes excluded? How could a 2 h 50 min marathoner be considered of ‘national level’ ?
- When during the day were the placebo or supplement taken?
- VO2max values are reported in table 1 without any mention of its measurement in the methods. Please explain the procedure.
- Line 120. Were the placebo and supplement similar in flavor, texture and appearance?
- Did you assess compliance for placebo and supplement intake?
- Line 126. Why no female included?
- Line 145. Measurement of PTH should be excluded or the rationale for its measurement explained.
Statistical analyses
- Line 162-166. Description of statistical analysis is complicated. A two-way anova (time x treatment) with a between subject factor should be used. And a post-hoc should be used to compare differences within and between individuals.
- Line 170. Please explain this method by Hopkins, while including the different formulas and steps.
- Line 172. Please explain the technique by Hopkins?
- Did you measure normality of distribution?
Results
- Line 179. There likely was a problem with the supplementation of the placebo and vitamin D3, or something changed drastically over the 2-week supplementation period among runners (maybe training or diet related or a lack of compliance with the intake of the vitamin or placebo). Indeed, in the control group pre-run levels of 25 (OH) D decreased in comparison to the value measured 2 weeks before whereas in the vitamin D3 group there was no increase in 25 (OH) D, despite daily high-dose supplementation. This represents a major flaw. Indeed, over the 2-week period levels should have been minimally maintained in the control group and increased in the vitamin D3 group. With no such observations, the credibility and validity of findings are severely affected. Indeed, it is not possible to ascribe the findings to the changes in 25 (OH) D, as there were none immediately prior to the run in the vitamin D3 group and were within the normal range for the placebo group.
- The values regarding completion time for the 100 km race need to be added. It would give a better idea of the levels of the runners to the reader and allow another comparison between the groups. If a difference exists between performances despite the similar VO2max, the relative intensity would be different between the groups and this could represent a potential bias.
- Table 2 and 3 could be combined.
Discussion
- As acute (Badenhorst et al., 2015; Eur J Appl Physiol) and chronic (McKay et al., 2019, Med Sci Sports Exerc) CHO restriction are known to affect IL-6 and hepcidin levels post exercise, the runners’ diet before the run may be discussed as a potential confounding variable. In this regard, the absence of dietary journal could be seen as a limit of the study.
- Line 310 - The sample size is stated as a limit of the study. Were any calculations done before the study to determine a minimal sample size?
- Line 312 – The time to complete the 100 km should be reported in the results so the reader can have a better idea of how ‘extreme’ was the effort.
Reviewer 2 Report
Dear authors, I suggest to check the text and the number e verify all the tables and numbers, the information at moment are not clear and the number not correct
The authors of paper “The Effect of Vitamin D3 Supplementation on Hepcidin, Iron and IL-6 Responses after a 100 km Ultra-marathon”, have studied the effect of supplementation of Vitamin D for two weeks in subjects before the 100 Km ultramarathon. The authors findings that the supplementation did not give influence on IL-6 and hepcidin after run but influence post exercise iron concentrations so one results is that vitamin D may affect iron regulation and erythropoiesis. This paper has a strong limitation: the relatively small group size.
Some suggestion are:
Row 134: before the run (pre), after 100 km (post) and 12 hours after the run (post12)
Comment: it would be useful to keep the same abbreviations also in the tables n.2 and n.3 because my questions are in relation with the table but also often on the text; here about the table:Table 2:Pre= before runPost = immediately after 100 kmTable 3:After the run = is the same … immediately after 100 km ?After the rest = after 12h rest If the venous blood sample are three, “post” and “after the run” are the same.The number about iron, hepcidin, ferritin, UIBC, TIBC and sTfR on table n.3 “after the run” could be the same of “post” in table n. 2?They are different, I do not understand if they derive from other analysis Row 179-180: level 25(OH)D “before”? And “after supplementation”? What means? Two more venous blood sample? Or before is one more venous blood sample and after supplementation is before the run? Row 272: why the group have not used only one single dose before the run, as Bacchetta as shown? Table 2 and 3: About table 2 and 3, could be interesting have the significative results about pre run (one more column after (pre)); the significative results after the run(one more column after (post)), and he same on table 3; all this could be more clear because on text the authors write about the significative results and on row 254 the authors write: …we did not notice a dissimilarity in hepcidin levels between VD and CON before the run: which is the statistics?
Reviewer 3 Report
This manuscript by Kasprowicz et al. focused on the relation among vitamin D, hepcidin, iron, and IL-6. Authors examined whether baseline vitamin D3 levels have any impact on post-exercise serum hepcidin, IL-6, and iron responses in 100-km ultra-marathon runners. There were no differences between the experimental (VD) and control (CON) groups before and after the run in the circulating hepcidin and IL-6 levels, but the decrease in iron concentration after the 100-km ultra-marathon was different between the groups. Authors concluded that various vitamin D3 status can affect the post-exercise metabolism of serum iron. Inflammation and anemia by the metabolism and complicated interaction among these factors are important in not only runners but also general population. Therefore, the concept of this study is understandable and results seem reasonable. Although overall manuscript seems written very well, authors may want to resolve several issues as below.
Major comments;
1) It is difficult to understand for readers the main message from this study what we should do before running. Minor comments;
1) Values of means and standard deviations seem better to be described with the first decimal places.
2) Conclusion of main text is too long. Most sentence in conclusions should be placed in the Discussion.